# Redox Changes during the Past 100 ka in the Deeper Eastern Arabian Sea: A Study Based on Trace Elements and Multivariate Statistical Analysis

**Ishfaq Ahmad Mir** [1,2,*]  **and Maria Brenda Luzia Mascarenhas** [1,*]

1    CSIR-National Institute of Oceanography, Dona Paula 403004, India
2    Geological Survey of India, SUKG, Bengaluru 560111, India
*     Correspondence: geoishfaq@gmail.com (I.A.M.); mariab@nio.org (M.B.L.M.)

**Abstract:** The temporal distribution of trace elements in a sediment core (SK117/GC-08) indicates minor changes in oxygenation during the last 100 kilo years in the bottom waters of the deeper eastern Arabian Sea. The high values of Mn, Co, Cu, Mn/Al, Co/Al, Cu/Al, V/Cr, and V/(V + Ni) in the sediments during interglacials and interstitials collectively indicate oxic conditions during warm periods. The high values of Cr, Ni, V, Mo, Cr/Al, Ni/Al, and Ni/Co in sediments during glacials and stadials collectively indicate dysoxic to suboxic conditions during the colder last glacial maximum and during the entire marine isotope stage two. The bottom waters have never experienced anoxic conditions. Multivariate statistics showed the attribution of the trace elements in two factors: cluster 1 (Co, Cu, Mn) was enriched during oxic conditions and cluster 2 (Cr, Mo, Ni, V) was enriched during dysoxic to suboxic conditions. Oxygenation conditions are mainly driven by variations in monsoon-controlled surface water productivity and changes in the flux of circumpolar deep water. The dysoxic to suboxic bottom water conditions developed at the core location during colder climates are very well synchronised with an increased organic matter flux. The main factor that controls the accumulation of the organic-rich sediments in the eastern Arabian Sea during a glacial is the increase in the supply of organic matter from increased primary productivity in the surface waters, controlled by winter monsoon winds, and localized convective mixing. During warmer interglacials and interstadials, the core location has remained well ventilated.

**Keywords:** marine sediments; geochemistry; Arabian Sea; late Quaternary; oxygenation; productivity



## 1. Introduction

Sedimentary archives from the Indian Ocean have provided lithogenic, biogenic, geochemical, isotopic, and granulometric evidence for variations in the Indian monsoon climate during the Quaternary period and its impact on the hydrography of surface waters, the flux of organic matter, variations in the thickness of the oxygen minimum zone (OMZ), and bottom water ventilation changes [1–5]. During the Quaternary period, the changes in the intensity and extent of the OMZ [2,6] in the intermediate water mass of the Arabian Sea (AS) region are directly linked with a variation in the surface water productivity and flux of organic matter [4,7,8]. The AS surface water increased productivity correlates with the low oxygen levels and shallow winter mixing driven by south west monsoon (SWM) winds and, vice versa, by north east monsoon (NEM) winds [9–11]. Fluctuations in the surface water productivity and OMZ in this region suggest strong linkages with the climate of higher latitudes [1,12]. The movement of heat, minerals, carbon, and oxygen throughout the world's oceans is significantly influenced by ocean circulation [13]. The North Atlantic and Southern Oceans are primarily where the deep and bottom seas entering the Indian and Pacific Oceans originate. When Antarctic Bottom Water (AABW) and North Atlantic Deep Water (NADW) are combined and changed by circumpolar currents, Circumpolar Deep Water is created (CDW). From circumpolar waters that upwell in the Indian and

Pacific Seas come the deep and bottom waters. The surface and intermediary waters from this upwelled water return to the North Atlantic Basin [13]. Two different water mass components make up the CDW. The term "Upper CDW" refers to a water mass between 2000 and 3800 m, which contains a significant amount of modified aged NADW, and the term "Lower CDW," also known as "bottom waters," refers to a water mass below 3800 m. Additionally affected by the Pacific Deep Water (PDW) and Indian Deep Water (IDW), the UCDW has lower salinity and lower oxygen levels than the LCDW [13]. LCDW, which joins the Indian Ocean through a Deep Western Boundary Current (DWBC) and ventilates the Arabian Sea basin via the Somalia Basin through the Owen Fracture Zone, is the main contributor to the bottom water mass in the Arabian Sea [13]. Proxy records also suggest changes in the AS are linked with deep water circulation [14]. Distribution of benthic foraminifera and preservation of organic matter at deeper depths of the AS are also influenced by the input of circumpolar deep waters (CDW). Oxygen contents of CDW are increased by the flux of North Atlantic Deep Waters (NADWs) [1,15]. In the Atlantic Ocean (AO), the flux of NADWs is high during interglacials and low during glacials [1,16]. The entry of deep water into the AS on glacial-interglacial timescales is controlled by the global sea level [5,17]. High oxygenated deep water fluxes are expected during interglacial and low during glacial periods.

Deep water circulation history is provided by the benthic microfossil and geochemical proxies. In this region, there are limited studies on the oxygenation history and its paleo-ceanographic significance. Geochemical studies of the sediment cores from deeper depths of the AS are required to understand the paleoenvironmental and paleoceanographic changes. Marine geochemistry is an important proxy to reconstruct the role of climate in the development of the redox state of bottom water [4,5,18,19]. The relative abundance of some trace elements such as Mn, Co, Cr, Cu, V, Ni, and their ratios are identified to vary with the bottom water conditions. The trace element distribution is largely controlled by the redox conditions and is a reliable proxy for reconstructing the palaeo-depositional conditions [5,20–23]. The oxic to anoxic conditions can be deciphered through the relative changes in the concentrations of some trace elements as they get enriched in the oxidizing zone and mobilized during the reducing conditions [24]. Contrastingly, some trace elements are depleted in the oxic zone [25–27]. Mn is enriched in the oxic zone and in reducing conditions is transferred to the pore waters, hence, gets depleted in the solid phase [28]. On the other hand, Cr, Mo, Ni, and V show the opposite behaviour and get depleted in oxic conditions and enriched in reducing conditions [24]. The anti-correlation is shown in the down core trend between Mn versus Cr, Mo, Ni, and V. Mo is a sensitive trace element, making it an excellent proxy to reconstruct the redox changes in the bottom water marine environment. Mo is highly enriched in reducing conditions in comparison with the other redox-sensitive trace elements [29].

During the glacial and interglacial periods, there have been major changes in ocean circulation. The export productivity and ventilation of the abyssal water mass play a major role in controlling the redox condition at the sediment-water contact. It is still unknown how bottom water circulation in the Indian Ocean affects variations in the redox state [30]. Seawater's redox state is typically classified as oxic (>2 mL $O_2$/L), dysoxic (2–0.2 mL $O_2$/L), suboxic (0.2–0 mL $O_2$/L), or anoxic (0 mL $O_2$/L) [13,30]. The suboxidized zone (very low $O_2$) and subreduced zone (no $O_2$ or $H_2S$), characterised by the $NO_3/N_2$ couple and $SO_4/H_2S$ couple, respectively, are two divisions of the suboxic zone [31]. Since various water masses contain varying amounts of oxygen, variations in their contribution have an impact on the redox state of the water mass. Less oxygenated UCDWs are found in the Antarctic Circumpolar Deep Waters (CDWs), which are located on top of more oxygenated LCDWs [13,27]. While the LCDW made up of modified Antarctic Bottom Water (AABW) is oxygen-rich (>4 millilitre $O_2$/L), the modified North Atlantic Deep Water (mNADW) is less oxygenated (1–4 mL $O_2$/L) and more silica-poor [3,13,27,30,32,33].

The solubility of trace elements such as V, U, Mo, Mn, Ni, Cu, and Cr as well as their redox state are controlled by the redox conditions at the depositional location [34].

Redox conditions at the depositional state are reflected by variations in the concentration of redox-sensitive elements. As a result, they can be used as a stand-in for research on paleo-redox conditions at the sediment-water interface and to deduce historical changes to the bottom water circulation. In this study we present information on trace elements (Co, Cr, Cu, Mn, Mo, Ni, and V) and their ratios in a sediment core from the deeper EAS covering the last 100 ka. The objective is to unravel the oxygenation changes and their impact on the paleoenvironment and paleoceanography in a deep-sea environment. In the deeper eastern AS (EAS), there are limited sediment geochemistry studies in comparison to the western AS (WAS). Trace element composition of the sediments from the EAS has varied in response to climate-dependent changes [34,35]. In this study, an attempt is made to fill the gap of pale redox changes in the deeper EAS using geochemical proxies.

## 2. Materials and Methods

A 408 cm long sediment core (SK117-GC08) was collected from 2500 m water depth in the EAS during the 117th cruise of research vessel Sagar Kanya from the location having geographic coordinates of latitude 15°29′71″ N and longitude 71°00′98″ E (Figure 1). The depth–age model for this sediment core is reported earlier [36]. The sediment core was subsampled at 2 cm intervals and a total of 100 samples are used in this study. The core preserves a continuous record of sedimentation for the past 100 ka. The sedimentation rate in the last 100 ka has varied from 2.3 to 6.1 cm/ka; the highest sedimentation has occurred during marine isotope stage 2 (MIS) [37]. Geochemical data were obtained by digesting the sediments using high-purity chemicals. The 50 mg oven-dried and salt-free sub-samples were subjected to open-PTFE vessel digestion on a hotplate in presence of a 10 mL mixture of $HF + HNO_3 + HClO_4$ (7:3:1) [33]. To make the final volume, the digest was dissolved in 4 mL of 1:1 $HNO_3$. The sample solution was analysed for major and trace elements using ICP-MS, Thermo X Series 2 (Plasma RF power) facility at CSIR-National Institute of Oceanography, Goa. Before running the samples, the analyst auto-tuned the instrument for the interested elements achieving satisfactory counts of the certified elemental standard solution. The precision of the results based on a few duplicate sample solutions and the accuracy based on two reference standards (MAG-1 and SGR-1) are within ±6%. Aluminium along with a few trace elements and their ratios are investigated in this study to understand the deep water circulation and surface water productivity controlled oxygenation changes in the bottom water of the EAS during the past 100 ka. Multivariate statistical analysis was carried out by using Origin Pro 2016 and MS Excel software. The results obtained by cluster analysis (CA) are shown by a dendrogram where the distance axis represents the degree of association between the trace elements. Higher values on the axis signify a weak association while the lower values signify a strong association between the variables. The dendrogram is derived from the hierarchical cluster analysis, the cluster method used is the nearest neighbour, and the distance type is an absolute correlation.

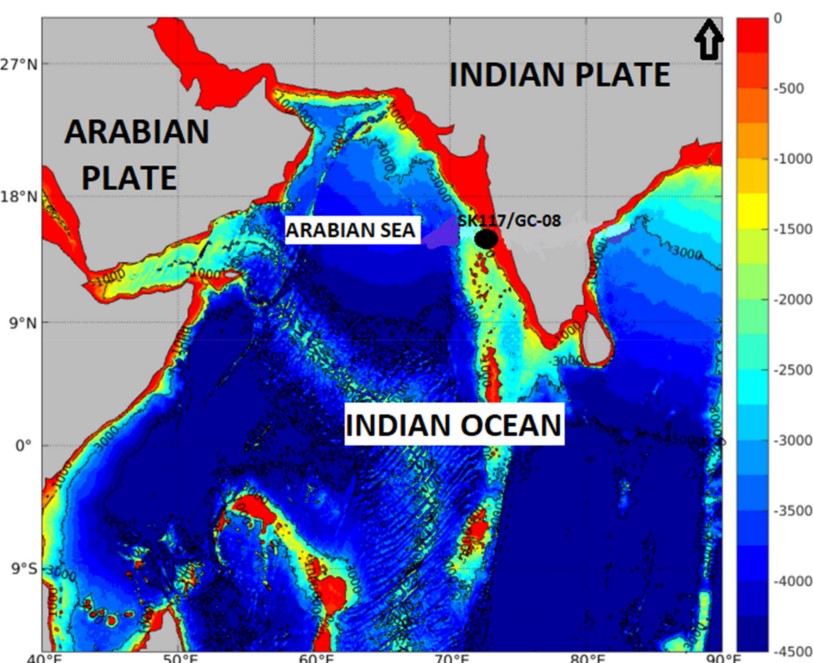

**Figure 1.** Location of the sediment core is shown in the eastern Arabian Sea in the northern Indian Ocean (SK-117/GC-08).

## 3. Results

*Trace Element Geochemistry*

The Co (ppm) varies from 14.61 to 29.32 with a mean of 19.18, Cr (ppm) varies from 93.30 to 192 with a mean of 132, Cu (ppm) varies from 36.73 to 82.72 with a mean of 52.41, Mn (ppm) varies from 488 to 1508 with a mean of 671.93, Mo (ppm) varies from 0.57 to 1.26 with a mean of 0.71, Ni (ppm) varies from 70.42 to 137 with a mean of 105, V (ppm) varies from 100 to 254 with a mean of 153, Cr/Al varies from 11.17 to 25.49 with a mean of 15.96, Co/Al varies from 1.38 to 3.59 with a mean of 2.33, Cu/Al varies from 4.00 to 10.37 with a mean of 6.34, Mn/Al varies from 51.18 to 158 with a mean of 81.30, Ni/Al varies from 8.40 to 18.89 with a mean of 12.69, Ni/Co varies from 2.40 to 7.35 with a mean of 5.55, V/Cr varies from 0.97 to 1.48 and has a mean of 1.16, and V/(V + Ni) varies from 0.52 to 0.69 and has a mean of 0.59. Co, Cu, Mn, Co/Al, Cu/Al, Mn/Al, V/Cr, and V/(V + Ni) values are high during warm periods (MIS 1, 3, 5a, and 5c) and Cr, Ni, V, Cr/Al, Ni/Al, and Ni/Co content are high during cold periods (MIS 2, 4, and 5b). Co, Cu, and Mn are strongly correlated with each other; likewise, Cr, Ni, and V are strongly correlated with each other.

## 4. Discussion

*4.1. Redox Changes Reflected in Trace Element Behaviour*

The changes in concentration and ratio of trace elements in the sediment core indicate the fluctuations in the oxygenation conditions in the bottom waters of the EAS. Ventilation changes in the marine sediments are inferred from the redox-sensitive trace elements [38] and their ratios [39]. Paleoredox changes can be identified by the fluctuations in the concentration of Mn in the marine sediments [40]. Under reducing conditions, Mn (IV) changes to Mn (II) in bottom waters and gets lost from the sediments to the water column; therefore, higher Mn values indicate the oxic depositional conditions [41]. In this study, Mn (ppm) and the ratio of Mn/Al (Figures 2 and 3) are high during warmer MIS 1, 3, 5a, and 5c and are low during colder MIS 2, 4, and 5b, suggesting more oxic conditions during interglacials enhanced the deposition of Mn. From the same sediment core, [7,8] reported that productivity in the EAS was high during colder glacial periods. The inverse relationship between Mn/Al and the productivity record of [7,8] confirm that the organic

matter flux controls the oxygen conditions of bottom waters. Rocks of peninsular India are depleted in Mn, hence the terrestrial Mn (ppm) supply to the deeper EAS by local river runoff is low [42]. Despite less supply of Mn (ppm) from the nearby landmass, there is an increase in the Mn/Al ratio of the sediment core during periods of decreased productivity [8]. This increasing Mn/Al ratio identifies the occasions when the reducing conditions in the sedimentary environment were low. During higher productivity, organic matter decomposition exhausts $O_2$ [12] and creates reducing conditions; therefore, the Mn/Al ratio becomes low. The ventilation changes reflected in the high Mn/Al ratio are controlled by the changes in the organic matter flux [43] to the sediments and consumption of $O_2$ during colder periods.

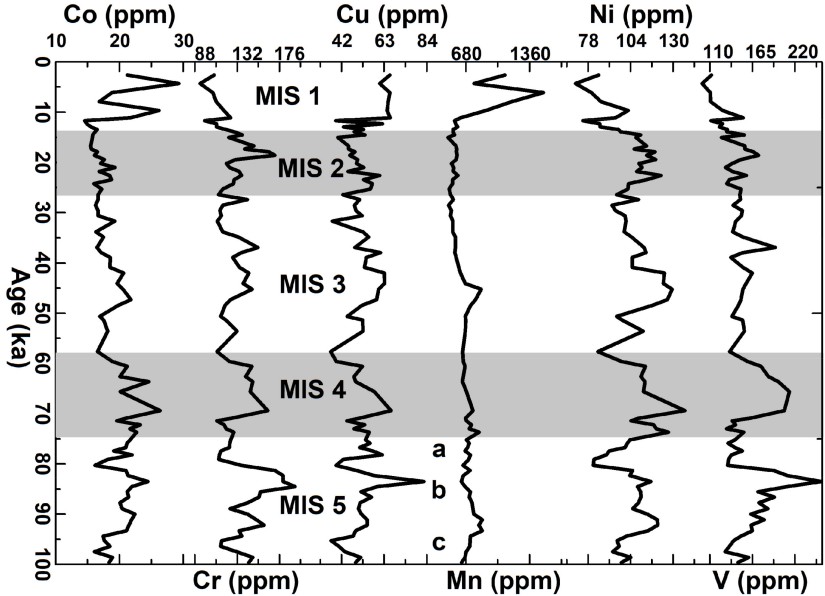

**Figure 2.** Down-core distribution of selected trace elements during the past 100 ka, in the sediment core SK117/GC08.

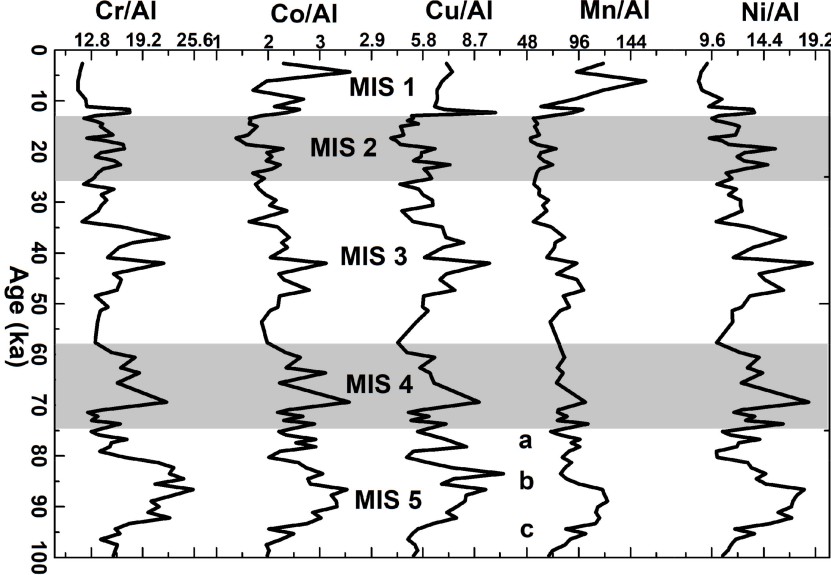

**Figure 3.** Down-core distribution of ratio of trace elements with respect to aluminium in the sediment core SK117/GC08 from eastern Arabian Sea during the past 100 ka.

Similarly, the concentration of Cr, Cu, Co, Ni, Mo, and V and their ratios are used as proxies for redox condition changes in bottom waters [24]. The down-core values of Co,

Cu, Co/Al, Cu/Al, V/Cr, and V/(V + Ni) are high during warm periods (MIS 1, 3, 5a, and 5c) and the down-core values of Cr, Mo, Ni, V, Cr/Al, Ni/Al, and Ni/Co content are high during cold periods (MIS 2, 4, and 5b) (Figures 2–4). The first group of trace elements shows a good correlation with each other; Co-Mn (0.41) and Co-Cu (0.50). Similarly, the second group also shows a good correlation with each other; Cr-Ni (0.59), Cr-V (0.82), and Ni-V (0.56). Mo (ppm) deposited under an anoxic marine environment preserves the redox conditions very well [33,44,45]. In a reducing environment, Mo (VI) is reduced to Mo (IV) and is precipitated from the solution [44]. The Mo (ppm) in the EAS core varies from 0.57 to 1.26 with a mean of 0.71 (Figure 5) peaking in MIS-2, especially during LGM. A Mo content between 5 to 40 (ppm) in the marine sediments indicates anoxic bottom water depositional conditions [29]. Our results are <5 (ppm), indicating oxic conditions with slight dysoxic to suboxic conditions during MIS-2. Mesozoic formations from different parts of the planet have very high Mo content (40 to 200 ppm) associated with high organic matter indicating strong anoxic conditions [33,45]. V has the tendency to become more concentrated in sediments underlying a suboxic to anoxic environment [39]. A ratio of V/Cr < 2 indicates oxic, 2 to 4.25 indicates dysoxic and >4.25 indicates suboxic and anoxic bottom water depositional conditions [46]. The V/Cr ratio in this study varies between 0.97 and 1.48 with an average of 1.16, suggesting the oxic bottom water environment. Ni/Co < 5 indicates an oxic environment, values between 5 to 7 indicate a dysoxic environment and >7 indicates a suboxic and anoxic environment [29]. The Ni/Co ratio of the sediment core varies from 2.40 to 7.35 with an average of 5.55, indicating that most of the time sediments have experienced oxic conditions except during MIS 2 when the sediments have experienced dysoxic to suboxic conditions, which is in accord with the changes in the productivity [8], flux of organic matter to the sediments, and consumption of $O_2$ during colder periods. The Cu/Al and Co/Al ratio are showing a similar trend down-core as that of Mn/Al, indicating enrichment during oxic depositional conditions of a warmer climate (Figure 3). However, Mo, Cr/Al, Ni/Al, and Ni/Co are showing an opposite trend down-core as that of Mn/Al, indicating enrichments during dysoxic to suboxic conditions of a colder climate (Figures 3–5). To further determine the redox conditions in the EAS during the past 100 ka, bivariate plots of V/Cr vs. Ni/Co and V/(V + Ni) vs. Ni/Co (Figure 6) are used [29]. In both of the bivariate plots, the data fall in the oxic to dysoxic zone. All proxies based on trace element parameters on average suggest the EAS core sediments have been deposited mostly beneath the oxic water column. The overall restricted range of the trace elemental concentrations indicates more or less stable oxic conditions [44]. The high productivity [8] induced dysoxic to suboxic conditions at the core location during MIS2. Similar observations are reported from the Arabian Sea earlier [47,48].

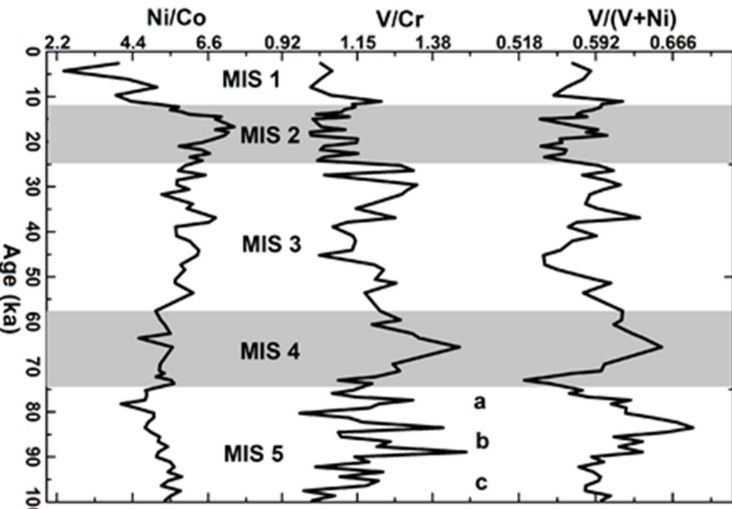

**Figure 4.** Down-core distribution of ratio of trace elements in the sediment core SK117/GC08 from eastern Arabian Sea during the past 100 ka.

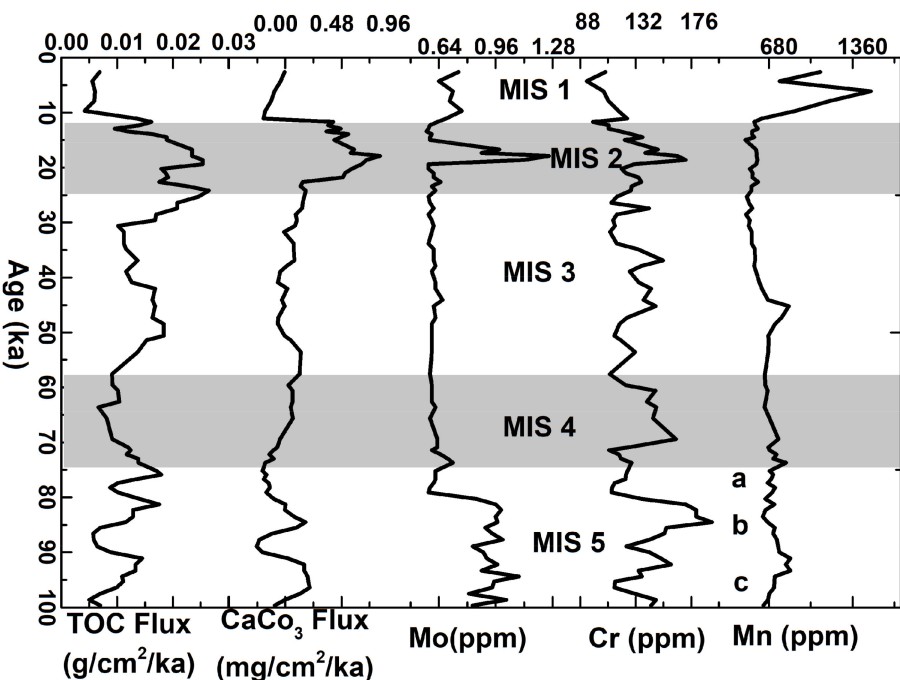

**Figure 5.** Down-core trend in the flux of TOC and CaCO3 [8], Mo and Cr and Mn during the past 100 ka in the eastern Arabian Sea.

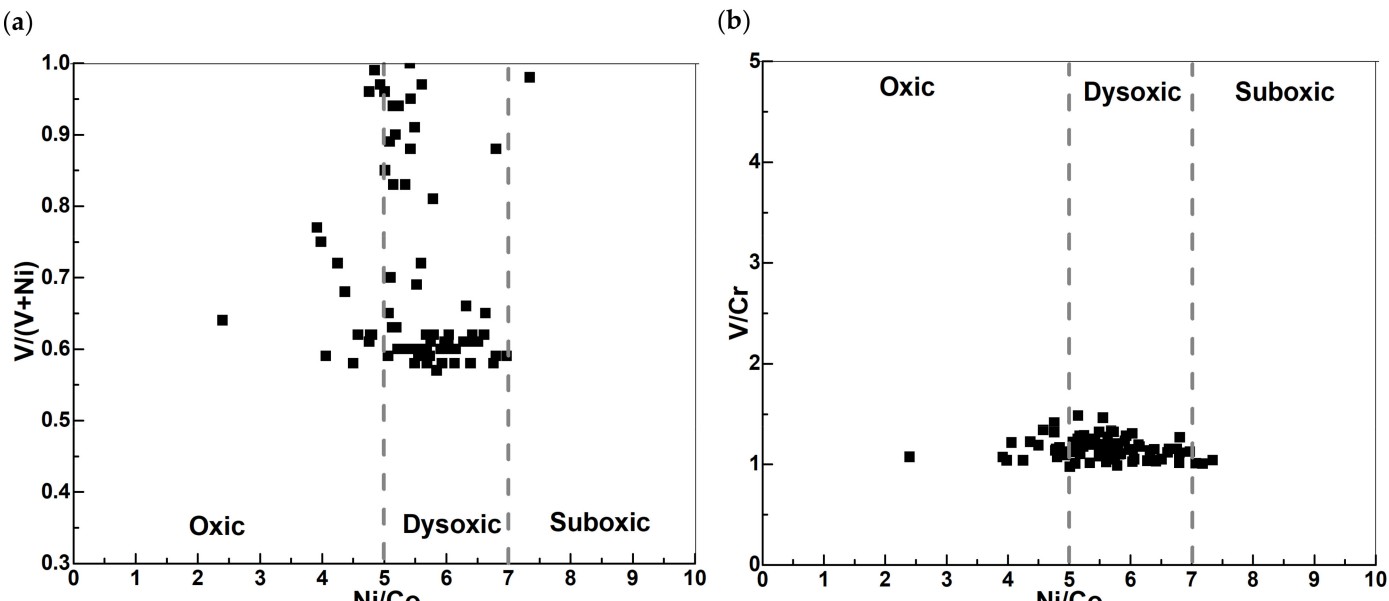

**Figure 6.** Bivariate plots of trace element ratios: (**a**) Ni/Co vs. V/(V + Ni); (**b**) Ni/Co vs. V/Cr, used as paleoredox proxies for the past 100 ka in the sediment core SK117/GC08. Boundaries between different redox conditions after Jones and Manning (1994).

*4.2. Statistical Methods Used in Reconstructing Behaviour of Redox Sensitive Elements*

This research used multivariate statistics to track how trace elements have changed over the past 100 ka in the eastern Arabian Sea, including hierarchical cluster analysis (HCA) and the Pearson correlation matrix (PCM). Without losing any of the crucial information, multivariate statistical analysis is able to describe the correlation between a large number of variables and condense the number of variables into a small number of factors [49]. When HCA and PCM are used together, marine sediments can be divided into different categories according to their redox chemistry characteristics [50]. Multiple

sedimentary environments have been effectively studied and classified using multivariate statistics [51]. The compositional trends and groups revealed by geochemical data obtained from geological samples can be used to deduce the sources and processes that led to the compositional changes. The process of building data and models is made simpler by the use of multivariate data analysis and statistical methods [52].

Multivariate statistical analysis of trace elements indicates two clusters (Figure 7a) of elements, cluster 1 (Co, Cu, Mn) and cluster 2 (Cr, V, Mo, Ni). The Pearson correlation matrix (Figure 7b) indicates Co, Cu, and Mn have a good correlation with each other; similarly, Cr, V, and Ni have a good correlation with each other. Two clusters of trace elements do not have a good correlation with each other. Inter-element relationships from multivariate statistical analysis provide the information on trace element sources, their behaviour in the depositional environment and pathways. The two cluster groups (Figure 7) are positively correlated within clusters and weakly to negatively correlate outside the clusters. Mn (ppm) indicates the deposition occurred under oxic bottom water conditions [41]. Co and Cu are positively correlated with Mn and are part of the same group, indicating oxic bottom water conditions. The enrichment of cluster 1 elements during interglacials and interstadials is due to oxic bottom water conditions, though the supply from the rocks of peninsular India is depleted [42]. On the other hand, Mo and V have the tendency to become more concentrated in sediments underlying suboxic to anoxic bottom water conditions [29,39]. The Mo, Ni, and Cr are positively correlated with V and are part of cluster 2, indicating dysoxic to suboxic bottom water conditions during glacials and stadials. The application of multivariate statistics showed the attribution of the trace elements in two factors: cluster 1 contains elements enriched during oxic conditions and cluster 2 contains elements enriched during dysoxic to suboxic conditions. Results obtained by applying the cluster analysis method are consistent with those obtained from Pearson correlation analysis.

(**a**)

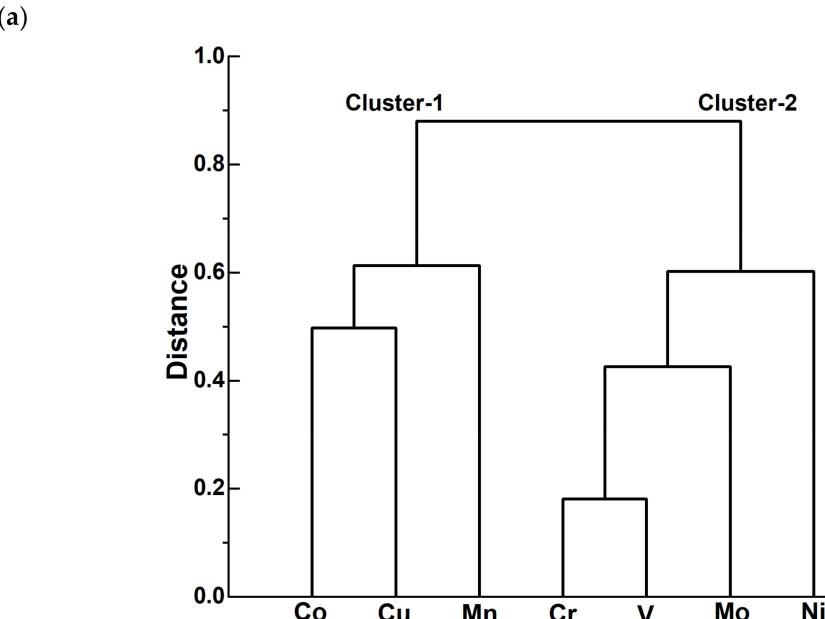

**Figure 7.** *Cont.*

(**b**)

| 2-Tailed Test of Significance | | Co | Cr | Cu | Mn | Ni | V | Mo |
|---|---|---|---|---|---|---|---|---|
| **Co** | Pearson Corr. | 1.00 | | | | | | |
| | *p*-value | -- | | | | | | |
| **Cr** | Pearson Corr. | 0.21 | 1.00 | | | | | |
| | *p*-value | 0.06 | -- | | | | | |
| **Cu** | Pearson Corr. | 0.50 | 0.23 | 1.00 | | | | |
| | *p*-value | 0.00 | 0.04 | -- | | | | |
| **Mn** | Pearson Corr. | 0.41 | −0.09 | 0.36 | 1.00 | | | |
| | *p*-value | 0.00 | 0.41 | 0.00 | -- | | | |
| **Ni** | Pearson Corr. | 0.29 | 0.59 | 0.24 | −0.11 | 1.00 | | |
| | *p*-value | 0.01 | 0.00 | 0.03 | 0.33 | -- | | |
| **V** | Pearson Corr. | 0.33 | 0.82 | 0.26 | −0.13 | 0.56 | 1.00 | |
| | *p*-value | 0.00 | 0.00 | 0.02 | 0.25 | 0.00 | -- | |
| **Mo** | Pearson Corr. | 0.07 | 0.61 | −0.04 | 0.18 | 0.16 | 0.42 | 1 |
| | *p*-value | 0.55 | 0.00 | 0.74 | 0.11 | 0.14 | 0.00 | -- |

**Figure 7.** Multivariate statistical analysis of trace elements (**a**) dendrogram derived from hierarchical cluster analysis, cluster method: nearest neighbour, and distance type: absolute correlation, (**b**) Pearson Correlation matrix and *p*-values using two-tailed test of significance.

Intensification and deepening of the OMZ in the AS and linked to the changes in orbital precession [53]. Bottom waters of the deeper EAS experience a sufficient inflow of oxygenated NADW during warmer climates, while there is a limited inflow of oxygenated NADW during colder climates which restricts the vertical expansion of OMZ [16]. The intensification and deepening of the OMZ in the EAS are decided by the coupled behaviour of high organic matter flux and reduced deep-water pumping [54]. The high flux of NADW and strong oxygenation of deep water are commonly confined to interglacial and interstadial conditions [14,55]. The flux of oxygen-rich deep-water and less productivity at the core site during interglacial and interstitial conditions prevented the development of anoxic conditions. However, the reduced flux of oxygen-rich deep-water and high productivity [56] at the core site during glacial and stadial conditions developed the dysoxic to suboxic conditions.

## 5. Conclusions

Paleoredox conditions in the EAS sediments based on trace element proxies suggest:

1. During the past 100 ka, bottom waters never became anoxic during the deposition of the sediments but became dysoxic to suboxic during the last glacial maximum (LGM) and MIS2. The changes in bottom water oxygenation are very well reflected in trace element content and their ratios.

2. The application of multivariate statistics on trace elements indicates two factors: the cluster 1 elements (Co, Cu, Mn) enriched during oxic conditions and the cluster 2 elements (Cr, Mo, Ni, V) enriched during dysoxic to suboxic conditions. Cluster analysis results are consistent with the Pearson correlation method.

3. Our results suggest that the oxygenation conditions in the deeper EAS during the past 100 ka are driven by variations in monsoon-controlled surface water productivity and changes in the flux of circumpolar deep water.

4. The dysoxic to suboxic bottom water conditions developed at the core location during colder climates are very well matched with increased organic matter flux. The high glacial productivity and weakening of ventilation are very well reflected in trace element proxies.

5. The major factor responsible for the accumulation of the organic material in the EAS during glacials was the high supply of organic matter to the core location due to increased surface water primary productivity driven by NEM and convective mixing. Dysoxic to suboxic bottom water conditions thus probably developed as a consequence of increased productivity, but the role of circumpolar deep water flux cannot be overruled.

**Author Contributions:** Conceptualization, I.A.M.; writing, I.A.M.; original draft, I.A.M.; methodology, I.A.M.; formal analysis, I.A.M.; writing—review and editing, M.B.L.M. All authors have read and agreed to the published version of the manuscript.

**Funding:** This research received no external funding.

**Data Availability Statement:** My submission does not include data in brief.

**Acknowledgments:** The authors are thankful to the director of CSIR-National Institute of Oceanography, Goa, India, for providing the laboratory facilities. The author wishes to thank V.K. Banakar (retired) for providing the sediment samples. Thanks are also due to J.N. Pattan (retired) for guiding I.A.M. for his Ph.D. degree. I.A.M is grateful to R.S. Garkhal, Geological Survey of India (GSI); J. Prasad, GSI, Hyderabad; D. Bhattacharya, GSI, Bengaluru, and A. Bhattacharya, GSI, Bengaluru, for providing the facilities to finalize this work. This is NIO contribution number 7039.

**Conflicts of Interest:** The authors declare that they have no known competing financial interests or personal relationships that could have appeared to influence the work reported in this paper. We the authors declare that this manuscript is original, has not been published before and is not currently being considered for publication elsewhere. The geochemical data used in the manuscript are from the Ph.D. thesis of the first author. We confirm that the manuscript has been read and approved by all named authors and that there are no other persons who satisfied the criteria for authorship. We further confirm that the order of authors listed in the manuscript has been approved by all of us. We understand that the corresponding author is the sole contact for the editorial process. He is responsible for communicating with the other authors about progress, submissions of revisions, and final approval of proofs.

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
