# Peer review of "Redox Changes during the Past 100 ka in the Deeper Eastern Arabian Sea: A Study Based on Trace Elements and Multivariate Statistical Analysis"

_water, doi:10.3390/w15071252_

Round 1

Reviewer 1 Report

Overview and general recommendation

            This work aims to unravel changes in oxygenation and their impact on the paleoenvironment and paleoceanography at the bottom of the eastern Arabian Sea.

Very interesting results are presented that complement other articles already published by the group.

However, several questions and suggestions need to be evaluated by the authors.

Major considerations

1. Lines 100 – 109. The authors bring the objective of the work. The text in addition to "The objective is to unravel the oxygenation changes and its impact on paleoenvironment and paleoceanography in deep-sea environment" is repeating information previously presented in the introduction. Review this paragraph.

2. What length of slices was the core cut?

3. Add the used ICP-MS settings.

4. Add the marks of the certified reference materials used.

5. Describe the statistical analysis used in the material and methods topic, not the discussion. Furthermore, Al concentrations should not be used in correlations. As can be evidenced in the presented Cluster (Figure 6), it is in a very distant group from the other elements studied in this work. Aluminum can be used as a normalizer.

6. Pearson's voice value alone didn't give us much information. Also put the significance test and the p-value.

7. Highlight the results presented in other works. Don't feature them again in your charts and tables. Cite that the results are from Mir et al., 2013.

8. Table 1 is not required. As well as the descriptive text of the results (between lines 165 and 177). I strongly advise authors to link the topical results and discussion. The journal allows this combination (see https://www.mdpi.com/journal/water/instructions).

9. I understand the relationship between organic matter and oxygen concentration. As well as the influence of environmental conditions on Mn nox. However, I cannot see in the Figures presented that in periods MIS 1, 3, 5a and 5c (this division of MIS 5 in the Figures is not clear) the concentration of Mn (and Mn/Al) is higher.

Minor considerations

10. Figure 1. Place an image of the sampling site with better resolution and/or in color.

11. Order the numbering of the figures.

12. Improve the quality (resolution) of the displayed figures.

Author Response

Dear Reviewer

Please find enclosed the comments addressed.

best regards

Ishfaq A. Mir

Reviewer 2 Report

The manuscript describes redox changes in 100 ka in the deeper eastern Arabian Sea using trace elements as proxies. Generally, the theme is interesting and the manuscript is written in good English, however, it would benefit from changes in the structure of the text. The text is not easy the follow and makes reading difficult. Primarily this refers to ‘Discussion’ part where the results presented in figures 2,3, and 4 are commented all together which is confusing. Furthermore, the order in which figures are shown in the text is not consistent – figure 5 appears after figures 6 and 7 – and figure 6 is missing.

Specific comments:

Line 108. Paleoredox?

Lines 249-251. Perhaps you should comment the data presented in one figure separately from the others and not all at once. This way you make a reader to jump between figures which makes reading difficult.

Line 256. There is no figure 6.

Line 293. Why is figure 5 mentioned after figures 6 and 7? Maybe you should number figures as they show in the text.

Figures are blurry. Please provide better quality figures.

Author Response

(The authors gave the same response as above.)

Round 2

Reviewer 1 Report

All questions were answered and suggestions were accepted. With the exception of ICP-MS settings. Show parameters like Plasma RF power, plasma gas flow, nebulizer gas flow, and auxiliary gas flow are essential for to method replicate.

Author Response

ICP-MS setting is Plasma RF power